# Impact of Incentive Policies and Other Socio-Economic Factors on Electric Vehicle Market Share: A Panel Data Analysis from the 20 Countries

**Chenlei Xue [1,2,\*]**, **Huaguo Zhou [3]**, **Qunqi Wu [2,4,\*]**, **Xueying Wu [1]** and **Xingbo Xu [1]**

[1] School of Transport Engineering, Chang'an University, Middle-Section of Nan'er Huan Road, Xi'an 710064, China; 2016023003@chd.edu.cn (X.W.); 2018023006@chd.edu.cn (X.X.)

[2] Center of Comprehensive Transportation Economic Management, Chang'an University, Xi'an 710064, China

[3] Department of Civil and Environmental Engineering, Auburn University, Auburn, AL 36830, USA; hhz0001@auburn.edu

[4] School of Economics and Management, Chang'an University, Middle-Section of Nan'er Huan Road, Xi'an 710064, China

\* Correspondence: xcl@chd.edu.cn (C.X.); wqq@chd.edu.cn (Q.W.)

**Abstract:** Under the strong support of policies and incentives, the global electric vehicle (EV) market has been developing rapidly. However, in the context of the overall EV market boom, the promotion policies and incentives for consumers to adopt EVs differ from country to country. It is worth exploring the key factors that affect market share and adoption of EVs, such as incentives, policies, and additional socio-economic factors. The data on EV market share and information on policies and incentives in 20 countries were collected from the published reports and online resources from 2015 to 2019. Random effects model analysis was conducted to explore the effect of various factors on EV market share. The innovation of this article is to combine incentive policies with socio-economic factors and use panel data to analyze the actual adoption behavior of the global EV market. Results show that the tax reduction policy, charger density, and income have significantly positive effects on the penetration of EVs. Thus, it suggested that government should still maintain tax incentives and focus on the deployment of charging infrastructure. Household income, as a socio-economic factor, also plays an important role in the adoption of EVs. This will help policymakers adjust and improve policy emphasis to promote the adoption of EVs.

**Keywords:** electric vehicles; incentive policies; socio-economic factors; random effects model

## 1. Introduction

Today, transportation accounts for nearly a quarter of direct $CO_2$ emissions which is a major contributor to air pollution. In order to reduce the emissions from motor vehicles, the transport sector sees the need to transform from conventional fuel vehicles to alternative fuel vehicles that are more environmentally friendly. The key to this shift is the electrification of transport modes, especially for electric vehicles (EVs). EVs can bring many environmental benefits over internal combustion engine vehicles by using one or more electric motors or traction motors for propulsion, which do not require gasoline fuel [1,2]. To decarbonize the transport system, governments around the world carrying out various incentive policies to foster the market deployment of this new, greener technology [3]. In some countries, the development of EVs has risen to the height of "national strategy" [4]. Ambitious policy announcements have been critical in stimulating the electric mobility transition in major vehicle markets. The sales of EVs worldwide have grown dramatically over the last decade, underpinned by supportive policies and technological advances. Global EV sales topped 2.1 million in 2019, surpassing the record year set in 2018, with EV stock reaching 7.2 million [5]. However, in the context of the tremendous sales growth, the uptake of EVs is still very limited in the majority of countries across the world although

the purported sales are increasing significantly. Furthermore, there was an uneven market share of EVs across the countries worldwide (see Figure 1). For example, in very few countries, such as Norway (55.93%), Iceland (22.60%), The Netherlands (15.14%), and Sweden (11.43%), the electric mobility fleet is expanding at a fast pace. The penetration rate of EVs was no more than 7% in the rest countries. Thus, it is necessary to explore the reasons for such a big difference.

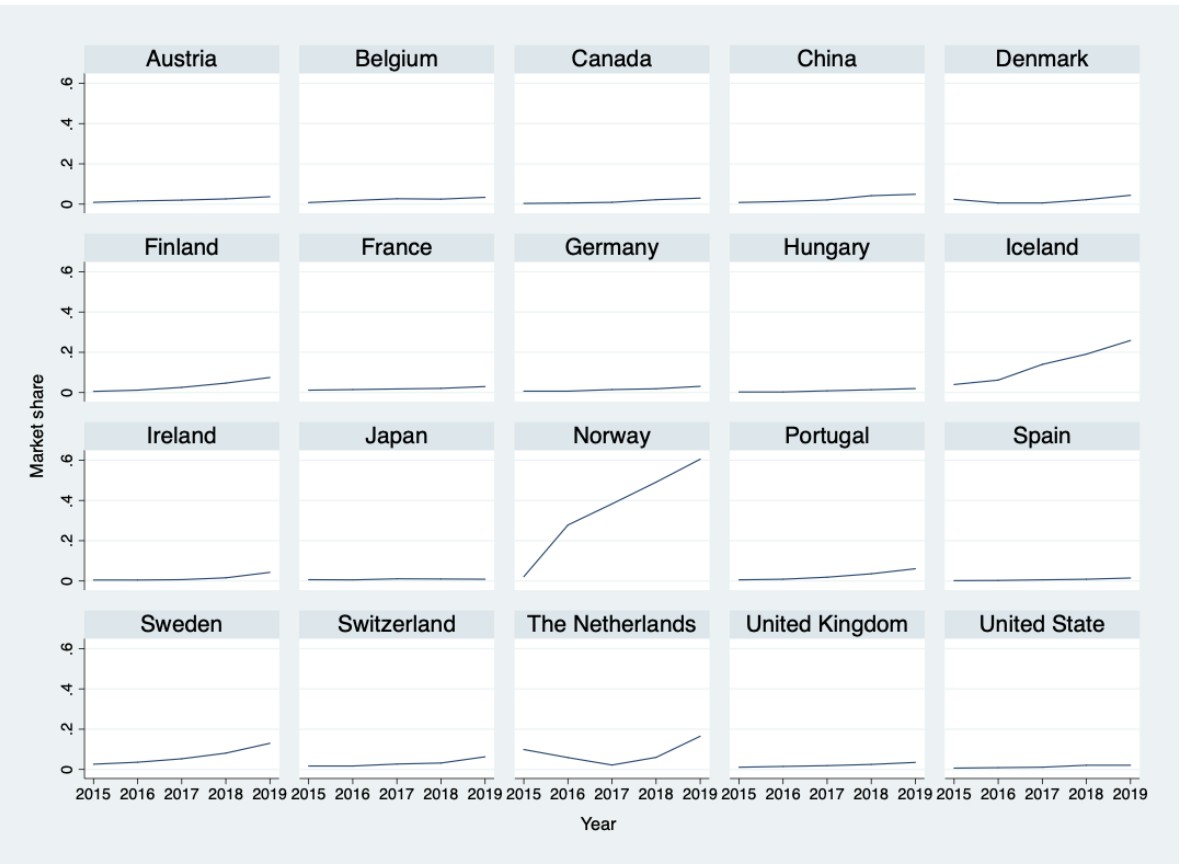

**Figure 1.** Time trend graph of 20 countries.

The objective of this paper is to better understand the factors that spurred the sales of EVs and evaluate the driving forces behind the success of the leading nations in EV market share. Specifically, it attempts to adopt a set of panel data from 20 countries over a period of 5 years to find out which factors have a significant positive impact on EV adoption through random effects regression model. The global stock of electric passenger cars reached 7.2 million in 2019, an increase of 40% year-on-year in 2019. China remains the world's largest EV market which 1.06 million EVs sold in 2019, followed by Europe (560,000) and the United States (326,000) [6]. Norway is the global leader in terms of EV market share (55.93% in 2019). The integrated data also reveal some general deployment patterns of the EV market in the world as shown in Figure 2. The analysis focused on the leading 20 EV markets (by share) that represented 90% of the world EV market based on the data from 2015 to 2019. Understanding the effect of the factors on the actual EV market in light of recent market developments will not only reveal the lessons learned by the leading countries in mobile electrification but also will help showcase a robust pathway for EV development for policymakers.

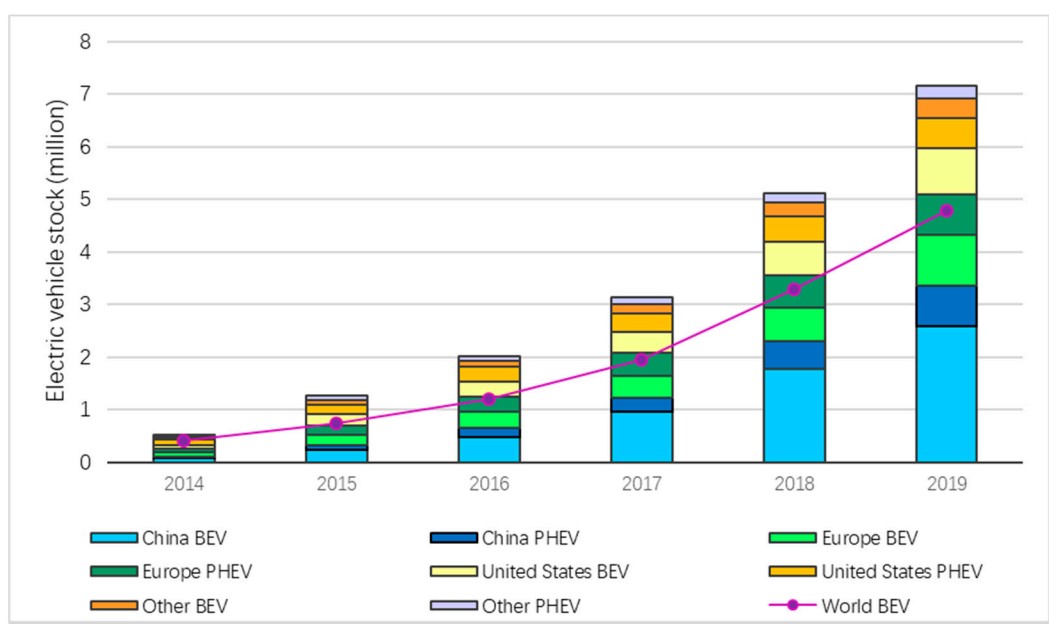

**Figure 2.** EVs stock in main markets, 2014–2019. SOURCE: Global EV outlook 2020 [5].

## 2. Literature Review

The worldwide EV market has grown since the 1990s [6]. By now, although the global electric cars exceeded 7.2 million in 2019, EVs only accounted for 2.6% of global vehicle sales and about 1% of global car stock [5]. The factors affecting the popularity of EVs can be summarized into three aspects: technical barriers, background factors, and individual characteristics. Specifically, the technical barriers include: driving range [7,8], performance [9,10], charging time [11], battery life [12], safety, and reliability [13,14]. Background factors: incentive policies [15,16], electricity prices and gasoline prices [17–19], and charging infrastructures [20]. Individual characteristics: socio-economic factors [21,22] and environmental awareness [23]. To stimulate EV adoption, a comprehensive package of incentive actions might help to overcome EV adoption barriers. Economic incentives and regulatory measures are often coupled with other policies that increase the value proposition of EVs. Considering various policy tools were having a major impact on the development of electric mobility. Many scholars have analyzed the impacts of past or current policy incentives implemented to promote EV adoption. They analyzed these factors from two perspectives: one is EV market analysis based on the actual adoption behavior, the other is through a questionnaire survey based on the consumer intention information. From the actual sales data study, most of this literature used the actual data to do quantitative research. Joeri H. Wesseling [24] compared the difference in plug-in electric vehicle policies across 13 countries from 2008 to 2014, the conclusion based on statistical analyses presented that infrastructure investments, sales incentives, and Research, Design, and Development (RD&D) subsidies were taken into considerations. Shafieia et al. [25] proposed the impact of different fiscal policy incentives on consumer decisions and the implications of fiscal incentives on overall economic benefits by using dynamic simulation modeling. Ning Wang et al. [16] found that chargers' density, fuel price, and road priority were significantly positive factors correlated with a country's EV market share and fiscal incentives were no longer the reasons for the differences in EV promotion among countries through utilizing multiple linear regression method. However, Sierzchula et al. [26] revealed that financial incentives, charging infrastructure, and local presence of production facilities are significantly positively correlated with a country's EV market share by using multiple linear regression analysis.

From the consumer's preference study, scholars used questionnaire results to predict which factors may influence consumers' willingness. For example, Gil and Nicholas [27]

studied that high occupancy toll stickers were more popular to the owners of plug-in hybrid electric vehicles (PHEVs) than the owners of battery electric vehicles (BEVs). Scott Hardman and Gil Tal [28] tried to know who are buyers of EVs and the importance of financial incentives to this group of adopters by an in-depth interview with 33 adapters. The result showed that environmental, performance, and technological motivations were the main reasons for adoption and financial purchase incentives were not important in consumers' decisions. Joseph S. Krupa et al. [29] administered a survey to 1000 residents in the US and found raising consumers' awareness of up-front incentives (e.g., purchase rebates) could have a greater impact than raising awareness of future fuel savings. Based on the stated-choice experiment, Joram H.M. Langbroek et al. [30] examined the effect of some potential policy incentives on EV-adoption including socio-psychological determinants. They concluded free parking and access to bus lanes were efficient incentives for the market breakthrough.

Based on the above literature review, it is clear to know that multiple policy approaches have shaped the configuration of the world's EV market today. However, many studies do not regard socio-economic factors as influencing factors in their studies [31,32], and most studies are contextualized in a single area or country with cross-sectional data. Some studies found that fuel and electricity prices, income may also lead to an increase in EV purchase [16,26]. In fact, due to "attitude-action gap", the information from the stated-choice survey may not be compliant with the actual purchase behavior of EVs [25]. Furthermore, as an emerging industry, the policy of EVs has been updated constantly according to the current market situation. Therefore, it is important to use multi-dimensional data to analyze the topic, or the result might not be able to reflect the current status quo of EVs. In general, depending on different periods and various measures of EVs, findings on the key factors for EV uptake differ significantly. Therefore, we combined several types of national policies with socio-economic factors in this study to yield insights for the future development and research of EVs.

## 3. Selected Incentives and Socio-Economic Factors

In the early stage of the development of EVs, countries around the world have introduced a series of incentive policies for switching to electric mobility. A stable policy framework and effective incentive mechanism are essential to creating a long-term and reliable market environment for EVs. However, the unprecedented "enthusiasm" in policy has not been transformed into hot sales in the terminal market. The mass popularization of EVs depends on individual users, while expanding the private consumer market is more difficult. Therefore, this study only considered passenger car sales due to most incentives for private consumers, not for fleet or business. Various types of measures were taken into consideration that may influence the decision of EV purchase indirectly. As discussed above, it can be categorized into three aspects: fiscal incentives, non-financial incentives, and socio-economic factors. Although the key to the scale-up of EVs is not just policy support, it played a pivotal role in transforming into an active adoption behavior.

### 3.1. Fiscal Incentive

Due to the high prices of EVs, there is a very small portion of consumers who actively chose to buy EVs. It is difficult for ordinary people to pay for such immature and expensive new things, especially when comparing with an internal combustion engine (ICE). Consequently, economic incentives can bridge the cost gap between EVs and less expensive ICEs. In light of this barrier, governments adopt a wide array of financial incentives to encourage EV use.

Although it has been proven that the financial incentives have a positive impact on EV adoption, there are some countries considering abrogating it or tax EVs. For example, China recently decided the subsidies will be completely withdrawn after 2020 [33]. In the US, 20 states implemented yearly fees on EVs, and more state legislators considered adding fees to EVs to make up the lost revenues from the fuel tax [34].

A fair amount of research conducted analyses of the effectiveness of monetary policies [35–40]. However, their findings pointed to the opposite. Some studies [35,36,38] found financial incentives did increase the rates of adoption for EVs. On the contrary, others [37–40] found there is no an approved relationship between certain economic incentives and EV adoption, but some other measures may increase the penetration of EVs.

### 3.2. Non-Financial Incentives

In general, fiscal incentives are often coupled with other non-financial incentives (e.g., waivers on fees, road priority, and access to restricted traffic) to attract more consumers, as these promotional actions can raise consumers' awareness of electric mobility. For example, EVs are waived on the parking fee or toll, allowed access to restricted traffic zones as road priority. It also helps increasing awareness of green vehicles on the roads. These types of non-financial incentives are very common in some leading markets, like in Norway [41], Sweden [30], and the US [42]. Additionally, charger density can directly reflect the strength of infrastructure incentives. While the advanced development of infrastructure is the key force to promote the development of EVs because it helps overcome range anxiety [43]. Unlike the fiscal incentive tool, not all countries provided non-financial incentives even though it is a focal point in EV policy discussions. Therefore, it is imperative to explore whether non-financial incentives have a positive impact on the adoption of EVs.

### 3.3. Socio-Economic Factors

For countries looking to become a prominent player in EVs, the implementation of any incentive measures is inseparable from the country's socio-economic background. As discussed above, many studies may only partially capture the impacts of financial incentives on EV adoption [44]. For instance, a past study [45] found that a 1% increase in gasoline price led to a 1.37% and 2.8% increase in PHEV and BEV purchases, respectively. Additionally, the level of income is supposed to be a factor to influence EV adoption [46,47]. As mentioned above, some studies [35,38] concluded that petrol fuel price is an important factor because it can lower the operating cost than ICEs. The cost of the usage phase like the electricity price should also be taken into consideration. It is nevertheless that incentive actions and socio-economic factors should not be considered separately from the organizational, economic, and societal context in which they perform a coordination function. Based on previous studies and the above discussions, Table 1 describes the dependent variable, independent variables, and sources of these data collected in this study.

**Table 1.** Description of variables and sources.

| Variables | Abbreviation | Description |
|---|---|---|
| Dependent variable | MS | National market share of EVs (numeric) [5,48] |
| Independent variables | PS | Purchase subsidies (numeric) [48] |
| | RT | Registration tax benefits (yes/no) [48,49] |
| | OT | Ownership tax benefits (yes/no) [48,49] |
| | VAT | Valued added tax benefits (yes/no) [48,49] |
| | CD | Charging density (yes/no) [48,50] |
| | RP | Road priority (yes/no) [48] |
| | WF | Waiver on fee (yes/no) [48] |
| | GP | Gasoline price (numeric) [51] |
| | EP | Household electricity price (numeric) [52] |
| | HI | Household disposal income (numeric) [53,54] |

## 4. Results and Discussion

### 4.1. Data Collection

As we mentioned in the former part, there are usually two approaches to analyze EV adoption. One is applying questionnaire data to explore the uptake of EVs, and the other is using actual sales data which is statistic data. For this paper, statistical data from

real adoption behavior is more meaningful to analyze actual EV adoption. There are three kinds of statistical data, and they are cross-sectional data, time-series data, and panel data. The data structure with three dimensions (individual, time, and metric) information is called panel data. The panel data can be used to construct and test more real behavior equations than the cross-sectional data or time-series data, which can be used for more in-depth and comprehensive analysis. We collected and analyzed the data from 20 countries with the leading electric vehicles market from 2015 to 2019. The data listed in Table 1 were collected from the world's top 20 countries with global market share leaders in EVs and data sources for this study. These countries collectively represented 90% of the total EV market share worldwide, including Norway, Iceland, Sweden, The Netherlands, Finland, China, Portugal, Switzerland, Austria, Belgium, United Kingdom, Denmark, Canada, France, United State, Germany, Ireland, Hungary, Japan, Spain. Since there were 25,210 fuel cell electric vehicles (FCEVs) worldwide which is significantly lower than the number of BEVs and PHEVs [55], and not only that, more than one-third of the global FCEVs is located in the United States, so in this paper EVs refer to battery electric cars as well as plug-in hybrid electric vehicles.

Based on the data availability, researchers collected data for the following variables from each country: EV market share, financial incentives (including purchase subsidies, registration tax benefits, ownership tax benefits, valued added tax benefits), non-financial incentives (including waivers on fees, road priority), household disposable income, gasoline price, electricity price, charger density. Since the tax benefits and non-financial incentives are treated as dummy variables in parts of the sample countries, which we can see in Table 2.

**Table 2.** Tax benefits and non-financial incentives in 20 countries.

| Countries | RT | OT | VAT | RP | WF |
|---|---|---|---|---|---|
| Norway | ✗ | ✓ | ✓ | ✓ | ✓ |
| Iceland | ✗ | ✓ | ✓ | ✓ | ✓ |
| The Netherlands | ✗ | ✓ | ✓ | ✓ | ✓ |
| Sweden | ✗ | ✓ | ✗ | ✗ | ✗ |
| Finland | ✓ | ✓ | ✗ | ✗ | ✗ |
| Portugal | ✓ | ✓ | ✓ | ✗ | ✓ |
| Switzerland | ✗ | ✓ | ✗ | ✗ | ✗ |
| China | ✓ | ✓ | ✗ | ✓ | ✓ |
| Austria | ✓ | ✓ | ✗ | ✗ | ✓ |
| United Kingdom | ✓ | ✓ | ✗ | ✗ | ✓ |
| Belgium | ✓ | ✓ | ✗ | ✗ | ✓ |
| Canada | ✓ | ✓ | ✗ | ✗ | ✓ |
| Denmark | ✓ | ✓ | ✗ | ✗ | ✓ |
| United State | ✓ | ✗ | ✗ | ✓ | ✗ |
| France | ✓ | ✓ | ✓ | ✗ | ✓ |
| Germany | ✗ | ✓ | ✗ | ✓ | ✓ |
| Ireland | ✓ | ✓ | ✗ | ✗ | ✓ |
| Hungary | ✓ | ✓ | ✗ | ✗ | ✓ |
| Spain | ✓ | ✓ | ✗ | ✗ | ✓ |
| Japan | ✓ | ✓ | ✗ | ✓ | ✗ |

### 4.2. Dependent Variables and Independent Variables

Table 3 lists the incentives and policy measures by 20 study countries. A descriptive analysis of fiscal incentives, non-financial incentives, and socio-economic factors was conducted. The results were summarized as below.

**Table 3.** Summary of random effects output.

| Dependent Variable | MS | |
|---|---|---|
| **Independent Variables** | **Co-Efficient** | ***t*-Statistic** |
| PS | −1.431 | 0.869 |
| RT | 0.142 | 0.805 |
| OT | 1.781 * | 0.070 |
| VAT | 0.179 | 0.715 |
| CD | 0.940 *** | 0.000 |
| RP | 0.182 | 0.668 |
| WF | −0.279 | 0.513 |
| GP | 0.001 | 0.997 |
| EP | −0.279 | 0.510 |
| HI | 2.157 *** | 0.000 |
| Observations | 100 | |
| R-squared | 0.665 | |
| Prob(F-statistic) | 0.0000 | |

Note: * significant at $\alpha$ = 10%; ** significant at $\alpha$ = 5%; *** significant at $\alpha$ = 1%.

### 4.2.1. Fiscal Measures

The determination of financial incentives was intricate because the approaches vary greatly from market to market: purchase subsidies, and tax benefits, or a combination of them. Purchase subsidies have the advantage of reducing the price of EVs. Tax reduction (registration tax benefits, ownership tax benefits, and valued added tax benefits) usually showed its benefits after three to five years. Thus, as this study focused on the national policies, purchase subsidies and tax benefits will be used to analyze fiscal policy. In order to ensure a relatively fair comparison, this paper uses the maximum purchase subsidy amount. Tax benefits are treated as dummy variables.

### 4.2.2. Non-Financial Measures

The financial incentives often can be offset by the price rise of EVs. The purchasing price is not the only decisive factor affecting the adoption of EVs. Many countries attempted to encourage EV adoption by combining financial incentives with non-financial incentives, such as waiver on toll and access to restricted traffic zones as road priority. All these non-financial incentives can reduce EV operational costs and enhance the awareness of green vehicles as well. Specifically, giving priority to EVs will also have an impact on other drivers for considering owning EVs. The charger density is a better indicator to reflect the level of charging infrastructure. Fast charging point, which could allow a long-distance trip, is crucial to the shift of electrification of transport since range anxiety is a big barrier to EVs. Normal charging point is an important supply for home and workplace charging. Therefore, in this paper, charger density includes the density of fast chargers and normal chargers.

### 4.2.3. Socio-Economic Factors

Previous studies found that national income per capita, gasoline prices, and electricity prices are closely related to the EV uptake. Household disposable income represents the real income level, which directly affects consumers' purchase decisions. At present, from the early stage of EV market to the gradually mature stage, the electricity prices are related to the operating cost of EVs that is another issue that people will consider. When comparing traditional fuel car, high gasoline price is a favorable factor for people to switch to EVs.

### 4.3. Panel Data Model

The sample data used in the panel model contains information from three dimensions: individual, index, and time. Set the dependent variable $Y$ and the explanatory

variable vector $X_{it} = (x_{1,it}, x_{2,it}, \ldots, x_{k,it})'$ in k-dimension. The standard static model with $i = 1, \ldots, N, t = 1, \ldots, T$ is

$$Y_{it} = \alpha_{it} + \beta_{it} X'_{it} + u_{it} \tag{1}$$

$X_{it}$ it is a K-dimensional vector of explanatory variables, without a constant term.
$\alpha_{it}$, the intercept, is independent of $i$ and $t$.
$\beta_{it}$, a (K × 1) vector, the slopes, is independent of i and $t$.
$u_{it}$, the error, varies over $i$ and $t$.

This study applied a panel regression model to get accurate regression results. Due to the different panel regression models have different processing forms for individual effects, panel regression analysis carries out pooled effects regression, fixed effects regression, and random effects regression, respectively. By examining the difference of the individual effects of the regression model, the regression form of the model is selected. F test was used to select between pool panel regression and fixed effects regression, then the Hausman test was conducted to decide whether the random effects model or fix effects model should be established. Through F test and Hausman test, a random effects regression model is established to test the relationship between EV market share and incentive policies as well as socio-economic factors. The analysis model specification is given as

$$lnY_{it} = \alpha_0 + \beta_1 PS_{it} + \beta_2 RT_{it} + \beta_3 OT_{it} + \beta_4 VAT_{it} + + \beta_5 lnCD_{it} + \beta_6 RP_{it}$$
$$+ \beta_7 WF_{it} + \beta_8 lnHI_{it} + \beta_9 lnGP_{it} + \beta_{10} lnEP_{it} + u_{it} \tag{2}$$

where: $i$ refers to a country; $t$ refers to the year. $Y$ refer to the dependent variable: EV market share, while the independent variables represented the purchase subsidy, registration tax benefit, ownership tax benefit, VAT benefit, road priority, waiver on fee, household disposable income, gasoline price, respectively. $u_{it}$ represents normally distributed random variable disturbance term. In addition, we took the logarithm of continuous variables as underlying variables to eliminate the heteroscedasticity in this paper [56]. We used Stata version 15.0 to estimate the analysis model.

## 5. Results and Discussion

Table 3 summarized the outputs of the random effects model fitting, where the R-squared is 0.665 which reflects the explanatory variable's explanatory degree to the explained variable reaches 66.5%. It can be proved that the model used in this study is of high goodness-of-fit. According to the result, the computed value of the analysis of *p*-value is 0.000 < 0.05, which means the analysis of this model is statistically significant, vice versa.

Table 3 summarizes the random effects model results. Since the regression model consists of different units and different scales, coefficients were used to analyze this model. The factors that produced significant results included ownership tax benefits, charger density, and household disposable income at $\alpha = 10\%$ and $\alpha = 1\%$, respectively. This means that they are the key factors for the national difference in EV adoption. Of these three variables, charger density, and household disposable income had a higher value of the coefficient, indicating it is the strongest at estimating adoption level. Among them, the effectiveness of ownership tax benefits (OT) was high (coefficient = 1.781), and this indicated that the implementation of this financial incentive will increase the uptake of EVs by 1.781%. The coefficient of household income (HI) and charger density (CD) were 2.157 and 0.940. This proved that household disposable income is indeed a strong influence. Specifically, for every 1% increase in household disposable income, the adoption of EVs surged by 2.157%. The coefficient of CD was 0.940, which means charger density had a positive effect on EV market share. If the government increases the density of charging points by 1%, it will stimulate the use rate of EVs by 0.9%.

## 6. Conclusions

This paper summarized the current market share, policies, and incentives on EVs in the world's leading 20 countries in the world. Further quantitative analysis of the

influence of various incentive policies and socio-economic factors on the EV market was conducted. The factors included in this study are purchase subsidies, registration tax benefits, ownership tax benefits, valued added tax benefits, charging density, road priority, waiver on fee, gasoline price, household electricity price, and household disposable income. The random effects model is conducted in this study, using the panel data of 20 countries from 2015 to 2019.

The results of the analysis suggest that ownership tax benefits, charger density, and income are key driving factors in all expanding EV markets. First, tax reductions, such as exemption from ownership, had a strong impact on the EV adoption, compared with purchase subsides. Although tax benefits do not give consumers direct savings at the beginning, they will decrease the overall operating cost during the operating stage. It suggests that, in the short term, governments still should undertake a substantial part of the spending on EVs through incentives schemes and tax waivers to trigger the mass market adoption. For example, due to the decline in sales, China announced that it will extend the fiscal subsidy originally planned to be eliminated in 2019 to 2022. [5]. Second, charger density could help overcome perceived and actual range barriers for EVs [46]. It should pay more attention to the deployment and construct enough chargers which is a prerequisite for wide adoption all over the world. The mass deployment of EV charging infrastructure plays a crucial role in the accessibility of chargers and EV electricity demand. In line with some previous research results, tax reduction and chargers are the two critical factors that have made a big difference in EV markets among the 20 countries. Third, household disposable income is one of the most neglected factors among socio-economic factors. The data analysis showed that household disposable income has the highest coefficient, which means income has a significant positive impact on EV uptake. As we all know, the price of EVs is higher than ICEs, which is one of the main reasons for the low market share in many countries. Although there are various incentives to alleviate the purchase cost, low income still would inhibit the popularization of EVs. This means that the government can formulate policies based on income levels to increase the possibility of electric vehicles popularization. A study from Sweden found that EV adopters usually have a strong purchasing power, and wealthier countries will adopt EVs earlier [8]. The information collected for this study showed that although some governments began to abrogate direct monetary subsidy for new EV purchases, the upfront cost of EVs and operating cost are still key decisive factors when compared with conventional cars.

There are three main suggestions for policymakers regarding future EV deployment. One, the existing financial incentives should not be eliminated in the short-term. The tax exemption is more effective than the direct purchase subsidies in the use phase of EVs. The governments still need to undertake the cost of EVs through incentive policies. Two, the deployment of charging infrastructure is a prerequisite for mass market adoption. The governments should expand the scale of charging points to ramp up density and it is a key measure to popularize EVs. High charger density can also increase the awareness of EVs, like roadside gas stations. Therefore, non-financial incentive measures can promote the adoption of EVs by raising consumers' awareness of EVs. Three, the governments should formulate policies based on income levels to adapt to different economic backgrounds. A combination of policies should be adopted worldwide, as it could accelerate the electrification of transport.

This study mainly focused on national-level incentive policies for EVs. It is suggested that the scope of EV policies should be the focus on some leading cities in future research. In addition, it has not been a long time to promote EVs in the global market, and only public data related to EVs in some major markets can be collected. Therefore, the sample size of this article needs to be increased.

**Author Contributions:** Conceptualization, C.X.; methodology, C.X.; software, C.X.; validation, C.X.; formal analysis, C.X.; writing-original draft, C.X.; writing-review and editing, C.X. and H.Z.; data curation, X.X.; visualization, X.W.; project administration, Q.W. and H.Z.; funding acquisition, Q.W. All authors have read and agreed to the published version of the manuscript.

**Funding:** This research was funded by the Strategic Planning Research Project of Ministry of Transport of China, grant number [2018-7-9] and [2018-16-9], Special Fund Project for Basic Scientific Research Operating Expenses of Central Universities (Humanities and Social Sciences Category) 300102239605, Shaanxi Provincial Social Science Fund (No.2020D028), Major Theoretical and Practical Problems Research Program Funded by Shaanxi Social Science Department (Program No. 2020Z362).

**Institutional Review Board Statement:** Not applicable.

**Informed Consent Statement:** Informed consent was obtained from all subjects involved in the study.

**Data Availability Statement:** The data can be collected from public websites and some annual reports. I quoted all data sources in this paper.

**Acknowledgments:** C.X. would thank my mentors, colleagues, and families. It was because of their support that I continued to improve this paper until it was accepted.

**Conflicts of Interest:** The authors declare no conflict of interest.

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
