# Peer review of "Impact of Incentive Policies and Other Socio-Economic Factors on Electric Vehicle Market Share: A Panel Data Analysis from the 20 Countries"

_sustainability, doi:10.3390/su13052928_

Round 1
Reviewer 1 Report
I don't think the manuscript has made many improvements compared to the last version.
I really appreciate the improvements the authors have made to this manuscript. However, I think the paper still has so many disadvantages that it may be inappropriate to be published by an academic journal with its current shape. That’s why I did give detailed comments this time.
Firstly, there are full of grammar errors. See Line 25, 105, 112, 125, 150, etc. Besides, many expressions are not in common use in English based on my knowledge. I do suggest a professional English editor to polish the manuscript.
Secondly, the methodology is problematic. The authors missed some pivotal factors which might influence the EV adoption, such as the household income, the annual travel distance, etc. I am concerned whether the conclusions are solid without taking these factors into account. Besides, it might not be appropriate to model the market share with the linear regression. Beta regression might be a better choice.
Reviewer 2 Report
The article presents an interesting analysis of the factors that can affect the market share of electric vehicles. The study is based on the survey of 20 countries that account for over 90% of the total market share of electric vehicles worldwide, including most of EU countries, China, the US and Japan.
The analysis took into consideration several measures that can affect the market share of electric vehicles, comparing them statistically. The analysis is conducted satisfactorily by highlighting the factors that contribute to incentivising the purchase of electric vehicles.
A slight criticism concerns the selection of the 20 countries (EU countries, China, Japan and USA) in which there is a very different use of private cars, if we consider the average distance travelled, the speed allowed and the availability and efficiency of the public transport, all factors that can influence the motivation to purchase electric vehicles. However, these factors not considered by this analysis can be examined in a further study.
Minor changes are needed as follows:
- The authors used the SPSS software for statistical analysis, so I suggest explicitly including the references. On the other hand, Figure 1 representing the SPSS user interface display of regression analysis in my opinion can be deleted.
- On Page 9 line 305 is reported “Person’s correlation coefficient (r)” instead of “Pearson’s correlation coefficient (r)”
Reviewer 3 Report
This interesting paper is almost based on quite simple statistical methods: Pearson’s correlation analysis and linear regression using OLS.
The statistical – econometric analysis presents a lot of approximations and need to be reexamined.
- Row 249 “The quantitative method allowed the researcher to collect numerical data”. No, this is not the appropriate way to explain what is quantitative method. Quantitative methods at the opposite of qualitative methods is based exclusively on quantitative variables and consequently required to collect the relative numerical data. The interpretation and conclusion drawn from the models are not quantitative methods. They are obligatorily parts of the analysis otherwise the results obtained have no interest. If quantitative method is based on data, qualitative method is based on words and meanings. Major qualitative methods concern interviews with open-ended questions, literature reviews exploring concepts and observations described in words.
Consequently, you cannot argue that your analysis is based on a mixed method. It is necessary to review the paragraph rows 248 to 253
- Row 256: correlation analysis, especially Pearson’s correlation does not in any case examine the influence of one or more independent variables on a dependent variable because it only analyzes the existence or not of dependency – association between two variables. In correlation analysis, you cannot suggest that one variable is dependent and the other independent contrarily with regression analysis. In any case, correlation implies. More over Pearson’s correlation examine only the linear association between two variables and not between one variable and another group of variables! Consequently, it is a measure of “apparent correlation” due to the fact it is possible that a third variable (or other variables) is related to both of the variables being investigated.
- Rows 262 -263, “The Pearson correlation coefficient is a parametric measure which can reveal the correlation between two variables”. You have to say: a parametric measure which can reveal the linear correlation between two variables.
- The paragraph 4.4. Analysis tool is absolutely not necessary, especially the 2 figures concerning SPSS. You have just to mention before paragraph 4.4. that the analysis has been implemented through SPSS (version 23).
- The correlation analysis has a very limited interest because as we can see most of independent variables have no significant linear association with your dependent variable MS. The question is then to which extent the independent variables present nonlinear association. You should have considered this point before using OLS that expressively requires linear association.
After table 4, it would be helpful for the reader to indicate what is the meaning of MS, TB, PS etc.
- The most important question in this text concern the specification of the model and the results in themselves.
Row 287, you say “?0 represents the error” , this is absolutely wrong. ?0 is the constant of the model and the true specification of the model is:
y=?0+?1?1+?2?2+?3?3+?4?4+?5?5+?6?6+?7?7+?8?1+?9? + ε
and you have to assume that the error term follows normal distribution.
The total number of observations is very small (20) so you have a degree of freedom very limited, contributing to increase the R2 but this is a statistical artifact. So when data are available, it is suggested to use panel data to increase the number of observations and obtained a more appropriate degree of freedom
Tabachnick and Fidell (2001) mention that OLS requires 20 “Subjects Per Variable” (SPV) and at least the minimum required should be 5 SPV. Another rule attributed to Harris is that the number of subjects should exceed the sum of 50 and the number of predictor variables. Harrell (2001) suggested that 10 SPV is the minimum required for linear regression models to ensure accurate prediction. When
Row 328 “It can be proved that the model used in this study is of high goodness-of-fit”
You say this because you obtain a R2 = 77.5% which is really high. BUT this result is probably not “true” because using OLS, you have to test if all the 7 hypotheses of linear regression are verified and the 1st one is the linearity of the model while it has already mentioned the importance of the degree of freedom.
As regards multicollinearity, you choose as threshold the value of 10 which is “accepted” by some researchers but not by all, especially with small samples. The best threshold value for VIF is 2.5 so you have at least 3 problems in terms of multicollinearity. You also did not present the Condition indexes of the Collinearity Diagnostics that are available with SPSS. This index is the most powerful for multicollinearity diagnosis.
What about Heteroscedacity? You mention (row 281) that you transform data through logarithm. Ok but it is not sufficient, you have to prove that there is no anymore problems.
Moreover, nothing is mentioned about the normality of errors.
Finally, even if you have a high R2, only 4 of the 9 «independent» variables are significant! This result is not surprising due to all the above mentioned.
Consequently, taking into account all the above, you will have to reconsider the interpretation of your model as well as your conclusions.
Two minor points:
Rows 191 – 193, the sentence starting with “In addition ...” is not clear.
Row 213, Footnote, “0 represents no this kind of incentive”, change it with “0 represents not having this kind of incentive”.
Row 353, “This paper summarized the current market share, policies and incentives on EVs in the 20 countries in 2019”. Mention that these 20 leader countries in the world.
Round 2
Reviewer 1 Report
The manuscript is improved somehow.
Reviewer 3 Report
All the requested improvements and corrections that I suggested have been done.
This manuscript is a resubmission of an earlier submission. The following is a list of the peer review reports and author responses from that submission.
Round 1
Reviewer 1 Report
A very small number of mistakes to be corrected. Maybe figure 1 could be improved by more clear left axe: 200 000 instead of 200000 etc.
Another issue - how will the author justify the difference to refere sometimes to 11 countries when market volume and share will be analyzed, and sometimes 20 countries included in the policy incentives analysis?
Author Response
Dear reviewer:
Thank you very much for your suggestions and your time. I have already modified my paper according to the comments.
Best regards and hope you have a good day.
Chenlei

Reviewer 2 Report
This paper tries to utilize the data from 20 countries to identify what factors influence the electric vehicles (EVs). The idea is good. However, the data analysis part is kind of problematic, some claims are messy, and there are many typos, leading to the unconvincing results. My comments are as follows.
The paper does not mention Figure 1 before its advent. You may want to refer to it. Besides, figure 1 shows the data of 2018, which however is never discussed before. If the 2018 data is available, why not use it in the later analysis?
In Figure 1, Table 2, Table 3, and all other locations showing data, please cite the data sources. It is very important for others to verify your research in the future.
Line 50, “quantity” should be “quantize”.
In Figure 1, I think it shows the “EV market sales and market share”.
Line 63, “The worldwide EVs market took off in 2013”: what do you mean “took off”? Do you have any idea support this claim?
Line 101, “may not compliant with” should be “may not be compliant with”.
Line 104, “the results might not able to” should be “the results might not be able to”.
Line 116, “socio-economic” what?
Line 129, based on the following analysis, it seems “charging station” and “household income” are not the fiscal incentives.
In Table 1, what year data is used here?
Line 172, “Since the fuel cell EVs (FCEVs) and hybrid electric vehicles (HEVs) without a plug have not yet entered the mass production stage for passenger vehicles, EVs refer to BEVs as well as PHEVs.”. I don’t agree with claim. At least, in the United States, HEV has been the major alternative fuel vehicle until now. See this study (Li et al., 2019)or check the website of “Alternative Fuels Data Center” of US. DOE to get more details.
What is the purpose of Table 2? It seems useless here. BTW, you are talking about Nissan Altima but showing Nissan Sentra in Table 2. They are different.
In Table 3, what “fees” do you mean? BTW, I think you are also taking about HOV lane before, but it is not shown here.
Line 197, “polies” should be “policies”.
Line 206, “HOV” lanes.
Line 207, you do not show the charging station information in Table 3.
You do not show the socio-economic factors either in any table. Actually, in Table 1, you should give a summary of all the variables. Check some crash modeling papers to get more details.
Table 5 and Table 6 can be merged together. Please highlight the significant coefficients in Table 6. Besides what is the standardized coefficient?
The model results in Table 6 seem good but not convincing. The beautiful goodness of fit in statistics does not necessarily mean the reasonableness in reality. For example, income has been proved as a major factor influencing EV adoption in many studies, but it is insignificant here. So, does it mean income would not influence the EV adoption? Currently, in US, the MSRP of a Tesla Model 3 starts from $33,815, and a Camry LE starts from $24,840. The price difference is so big that it obviously influences the Tesla adoption. How do you prove income is not important here?
Besides, EVSE (charging stations) is also insignificant. Firstly, I think the density of charging stations rather than the number of charging stations should be used here, as areas and populations of these countries are so different. Secondly, charging infrastructure is also thought to be able to influence EV adoption in many studies. However, based on your finding, it seems not important. So, are we going to continuously build the charging stations?
Similarly, for other variables, what practical suggestions can you give based on the estimated results? Are the estimated results really meaningful for practice?
Author Response
Dear reviewer:
Thank you very much for your suggestions and your time. I have already updated the data and modified my paper according to the comments. And I did some explanations for some comments below.
Best regards and I hope you have a good day.
Chenlei
1.You do not show the socio-economic factors either in any table. Actually, in Table 1, you should give a summary of all the variables. Check some crash modeling papers to get more details.
Table 1 shows the socio-economic factors and it contains the details
2.Table 5 and Table 6 can be merged together. Please highlight the significant coefficients in Table 6. Besides what is the standardized coefficient?
In regression analysis, different units and different scales are often used. For example, one variable might use dollars and another might use percentages. Standardizing coefficients means that you can compare the relative importance of each coefficient in a regression model.
3.The model results in Table 6 seem good but not convincing. The beautiful goodness of fit in statistics does not necessarily mean the reasonableness in reality. For example, income has been proved as a major factor influencing EV adoption in many studies, but it is insignificant here. So, does it mean income would not influence the EV adoption? Currently, in US, the MSRP of a Tesla Model 3 starts from $33,815, and a Camry LE starts from $24,840. The price difference is so big that it obviously influences the Tesla adoption. How do you prove income is not important here?
The monetization benefits of many incentives policy to consumers vary according to the technical level of the vehicle’s model, so we need to select a representative model and conduct a quantitative evaluation of consumer benefits according to the technical level of a representative model. In this paper, a top-selling EV model that can represent the choice of most people was selected to compare different incentives among 20 countries. Though not conforming to reality, this paper supposes that the selected vehicle models are available in all 30 selected countries and vehicle prices (excluding taxes and subsidies) are the same. This ensures a relatively fair comparison and assessment of incentives around the world. Only in this way, we can get a relatively objective result.
Reviewer 3 Report
The authors deal with interesting and relevant topic. However, there are several things that should be tackled and improved in order to make the paper appropriate for publication:
Contributions with respect to existing literature should be stated more clearly, whether they are related to comprehensive analysis including more factors, newer data, new insights, etc. A software used for calculating statistics should be stated.
How the data listed in Table 2 is relevant for the rest of the paper? Where these numbers are used? It would be helpful if numerical data used for correlation and regression analysis is listed in table, thus providing the possibility for repeating of the results.
Currently, the data is only partially given in Table 3, e.g. given for Purchase subsidy, Waiver on fees, Access to bus lanes, while not for Ownership tax and Registration tax. Also, the graphical analysis of this data, i.e. dependence of market share with respect to individual independent variables, would provide intuition and would further improve the paper.
TR from Table 4 is not defined or it is typographical error (i.e. there is no TR in Table 1). Intercorrelations between independent variables from Table 4 are not commented (only correlations with respect to dependent variable are commented).
It seems there is significant correlation between some independent variables, e.g. TR with respect to PS, which is in contradiction with the conclusion given related to data given in Table 6, where it is stated that there are no collinearity problems. Which method is used for calculating model (1) parameters? Least squares and normal equation?
Results related to regression model given in Table 5 seems to indicate potential overfitting problem, since there is significant difference between R2 and R2adj values. This could be caused by low amount of data examples (only 20), and relatively large number of model parameters (10). Please comment on this.
It is stated that ANOVA analysis confirms that the model is significant (above Table 5). However, this should be explained in more details, by refereeing to the numbers in Table 5. Generally, the data provided in tables should be explained with an emphasis on its utility for the paper subject, e.g. what is VIF and how it is shown that there is no problem with collinearity (what is threshold for VIF to conclude this)?
In the conclusion section it is stated that purchase subsidy has negative impact on the market share “Third, the data analysis showed that direct purchase subsidy policy has a negative effect on EVs 297 share.”, which is probably based on the correlation results given in the Table 4. On the contrary, in the regression analysis, the coefficient related to the purchase subsidy (PS) is positive (e.g. standardized equal to 0.178) meaning there is positive impact of PS on the market share, which contradicts result in the Table 4.
The authors should comment on this. Why in Table 6, the unstandardized coefficients are denoted as B, while the standardized ones are denoted as Beta?
Author Response
Dear reviewer:
Thank you very much for your suggestions and your time. I have already updated the data and modified my paper according to the comments. And I did some explanations for some comments below.
Best regards and I hope you have a good day.
Chenlei
1.Contributions with respect to existing literature should be stated more clearly, whether they are related to comprehensive analysis including more factors, newer data, new insights, etc. A software used for calculating statistics should be stated.
I have revised it in this paper.
2.How the data listed in Table 2 is relevant for the rest of the paper? Where these numbers are used? It would be helpful if numerical data used for correlation and regression analysis is listed in table, thus providing the possibility for repeating of the results.
The monetization benefits of many incentives policy to consumers vary according to the technical level of the vehicle’s model, so we need to select a representative model and conduct a quantitative evaluation of consumer benefits according to the technical level of a representative model. In this paper, a top-selling EV model that can represent the choice of most people was selected to compare different incentives among 20 countries. Though not conforming to reality, this paper supposes that the selected vehicle models are available in all 30 selected countries and vehicle prices (excluding taxes and subsidies) are the same. This ensures a relatively fair comparison and assessment of incentives around the world. Only in this way, we can get a relatively objective result.
3.Currently, the data is only partially given in Table 3, e.g. given for Purchase subsidy, Waiver on fees, Access to bus lanes, while not for Ownership tax and Registration tax. Also, the graphical analysis of this data, i.e. dependence of market share with respect to individual independent variables, would provide intuition and would further improve the paper.???
I have revised Table 3.
- seems there is significant correlation between some independent variables, e.g. TR with respect to PS, which is in contradiction with the conclusion given related to data given in Table 6, where it is stated that there are no collinearity problems. Which method is used for calculating model (1) parameters? Least squares and normal equation?
Correlation analysis mainly describes the close degree of quantitative correlation, while regression analysis focuses on the degree of influence of the change of independent variables on the dependent variables. The default parameter estimation method of SPSS software is the least square method. And I've updated the latest figures and the results have changed a bit.
5.Results related to regression model given in Table 5 seems to indicate potential overfitting problem, since there is significant difference between R2 and R2adj values. This could be caused by low amount of data examples (only 20), and relatively large number of model parameters (10). Please comment on this.
I haven't considered the problem you mentioned, because my major is not statistics. But I think there's some truth in what you're saying, and I'll think about it more in the future. Thank you.
6.It is stated that ANOVA analysis confirms that the model is significant (above Table 5). However, this should be explained in more details, by refereeing to the numbers in Table 5. Generally, the data provided in tables should be explained with an emphasis on its utility for the paper subject, e.g. what is VIF and how it is shown that there is no problem with collinearity (what is threshold for VIF to conclude this)?
I explained VIF in this paper.
7.In the conclusion section it is stated that purchase subsidy has negative impact on the market share “Third, the data analysis showed that direct purchase subsidy policy has a negative effect on EVs 297 share.”, which is probably based on the correlation results given in the Table 4. On the contrary, in the regression analysis, the coefficient related to the purchase subsidy (PS) is positive (e.g. standardized equal to 0.178) meaning there is positive impact of PS on the market share, which contradicts result in the Table 4.
I've updated the latest figures and the results have changed a bit. And the value of P (PS) is not less than 0.05.
8.The authors should comment on this. Why in Table 6, the unstandardized coefficients are denoted as B, while the standardized ones are denoted as Beta?
This is the system default setting for SPSS results
Round 2
Reviewer 2 Report
Not big improvements.
Reviewer 3 Report
The below are given some comments related to reviewer’s answers.
Comment related to Answer 1. I cannot find statement in the paper related to which software is used for conducting statistical analysis.
Comment related to Answer 4. For the sake of completeness this should be stated that the model parameters are calculated in SPSS software by using the least square method.
Comment related to Answer 5. R2adj index is very simple modification of R2 index (https://www.statisticshowto.datasciencecentral.com/adjusted-r2/), often used when there is low amount of data and relatively complex model, since in this case R2 index can be misleading. Potential problematic of this should be at least commented in one sentence for the sake of completeness. Otherwise, there is no sense to have it included in the table.
Comment related to Answer 7. Since it is counter-intuitive that purchase subsidy (PS) has negative effect on EV market share, here probably caused by some hidden/not considered variable/s which correlate well with PS, the authors should at least milden the corresponding conclusion or provide potential explanation why PS could have negative impact on EV market share.
Line 203: Some mistake. Only word Though.
In Table 5 it can be seen that correlation index R is 0.896. However, in the text above the table (line 319) it is stated that this index is 0.911.
In table 1, the abbreviation MS would be more appropriate for Market Share instead of MK.